# Continuous Electrical Stimulation Affects Initial Growth and Proliferation of Adipose-Derived Stem Cells

**DOI:** 10.3390/biomedicines8110482

**Published:** 2020-11-08

**Authors:** Peer W. Kämmerer, Vivien Engel, Franz Plocksties, Anika Jonitz-Heincke, Dirk Timmermann, Nadja Engel, Bernhard Frerich, Rainer Bader, Daniel G. E. Thiem, Anna Skorska, Robert David, Bilal Al-Nawas, Michael Dau

**Affiliations:** 1Department of Oral and Maxillofacial Surgery, Facial Plastic Surgery, University Medical Center Mainz, 55131 Mainz, Germany; daniel.thiem@unimedizin-mainz.de (D.G.E.T.); bilal.al-nawas@unimedizin-mainz.de (B.A.-N.); 2Department of Oral and Maxillofacial Surgery, Facial Plastic Surgery, University Medical Center Rostock, 18057 Rostock, Germany; vivien.engel@med.uni-rostock.de (V.E.); nadja.engel@med.uni-rostock.de (N.E.); bernhard.frerich@med.uni-rostock.de (B.F.); michael.dau@med.uni-rostock.de (M.D.); 3Institute of Applied Microelectronics and Computer Engineering, University of Rostock, 18051 Rostock, Germany; franz.plocksties@uni-rostock.de (F.P.); dirk.timmermann@uni-rostock.de (D.T.); 4Department of Orthopedics, University Medical Center Rostock, 18057 Rostock, Germany; anika.jonitz-heincke@med.uni-rostock.de (A.J.-H.); rainer.bader@med.uni-rostock.de (R.B.); 5Department of Cardiac Surgery, University Medical Center Rostock, 18059 Rostock, Germany; anna.skorska@med.uni-rostock.de (A.S.); robert.david@med.uni-rostock.de (R.D.); 6Department Life, Light & Matter (LL&M), University of Rostock, 18059 Rostock, Germany

**Keywords:** tissue engineering, human adipose-derived stem cells, proliferation, electrical stimulation, continuous stimulation, alternating current

## Abstract

The aim of the study was to establish electrical stimulation parameters in order to improve cell growth and viability of human adipose-derived stem cells (hADSC) when compared to non-stimulated cells in vitro. hADSC were exposed to continuous electrical stimulation with 1.7 V AC/20 Hz. After 24, 72 h and 7 days, cell number, cellular surface coverage and cell proliferation were assessed. In addition, cell cycle analysis was carried out after 3 and 7 days. After 24 h, no significant alterations were observed for stimulated cells. At day 3, stimulated cells showed a 4.5-fold increase in cell numbers, a 2.7-fold increase in cellular surface coverage and a significantly increased proliferation. Via cell cycle analysis, a significant increase in the G2/M phase was monitored for stimulated cells. Contrastingly, after 7 days, the non-stimulated group exhibited a 11-fold increase in cell numbers and a 4-fold increase in cellular surface coverage as well as a significant increase in cell proliferation. Moreover, the stimulated cells displayed a shift to the G1 and sub-G1 phase, indicating for metabolic arrest and apoptosis initiation. In accordance, continuous electrical stimulation of hADSC led to a significantly increased cell growth and proliferation after 3 days. However, longer stimulation periods such as 7 days caused an opposite result indicating initiation of apoptosis.

## 1. Introduction

For decades, the induction of cell proliferation has been a central component in the research of tissue regeneration, which is a complex process involving a diversity of cell types whose functions are regulated by intricate networks of biochemical signals. An important phase of regeneration is the proliferation of precursor cells. After a proliferative phase, precursor cells differentiate into tissue-specific cell types that will build up their characteristic extracellular matrix and regain their original function [1,2]. Mesenchymal stem cells are adult stem cells providing advantages of easy isolation, expansion and low antigenicity, allowing allogenic transplantation [3,4]. Although the allogenic stem cell therapy is discussed controversially from an ethical point of view, successes have already been achieved in various fields such as the therapy of neurodegenerative [5], cardiovascular [6] or musculoskeletal diseases [7,8]. Besides application of certain growth factors or the alteration of culture conditions to modulate stem cell differentiation [9,10,11], electrical stimulation is a well-known method used to induce changes in various cellular processes such as apoptosis, proliferation and cell differentiation [12,13,14]. It is assumed that the cells’ transmembrane potential and influx of calcium ions is altered, which may have a significant influence on function and cell metabolism [15]. In this context, the directional migration of cells in response to electrical stimulation has already been demonstrated [16,17]. The effect of electrical stimulation on cells is influenced by several dependent variables such as electrical field strength (V/m), frequency (Hz), the duration and modus of application (continuous versus pulsatile) [18,19,20,21] as well as the type and origin of the stimulated cells and the culture conditions [22]. Further on, electrical stimulation can be divided into mono- or biphasic and into different waveforms such as pulsed, sinusoidal, square and triangular [2]. Regarding stimulus frequency, electrical stimulation frequencies below 1 kHz have been shown to increase cell proliferation by affecting the cell cycle, increasing the cell fraction and synthesizing DNA [2,23].

Electrical stimulation might be suitable to increase osteoblast activity that plays a pivotal role for osseous reconstitution in case of fractures or defect reconstruction [24,25]. Comparably to this, an increase in the osteogenic differentiation of human adipose stem cells via electrical stimulation has been demonstrated, whereby the impact on cells differed depending on the type of electrical stimulation [14,22]. Dauben et al. demonstrated a voltage-dependent increased differentiation of human osteoblasts when applying alternating current (AC) sinusoidal signals with a frequency of 20 Hz [26]. Furthermore, cell experiments using frequencies above 1 kHz have been shown to increase cell differentiation when maintained at low intensity [27,28]. In order to predict the influence of electrical stimulation on the regeneration of bone, our group has already investigated and described numerical calculations as the design basis of an electro-stimulation system for in vivo use on mandibular critical size defects [29]. For this purpose, the defect to be stimulated has to be filled with bone forming stem or progenitor cells in order to restore the missing bone volume. The success of osseous regeneration largely depends on the cell type and adipose-derived stem cells are commonly studied for bone tissue engineering approaches due to their multipotency, ease of the collection and efficient culture conditions [22,30]. Therefore, the aim of our present study was to analyze and establish suitable parameters for continuous electrical stimulation in order to improve the growth, viability and proliferation of human adipose-derived stem cells in vitro.

## 2. Experimental Section

### 2.1. Cell Culture

Lipoaspirate samples were collected from patients who have undergone liposuction or lipofilling procedures at the University Medical Center Rostock, Germany, after approval from the local Ethics Committee (No. A 2014-0092). The procedure for isolating adipose-derived stem cells has already been described [31]. In brief, the samples were incubated with 6 mg/mL collagenase NB4 (SERVA Electrophoresis GmbH, Heidelberg, Germany) using a Lab Rotator at 37 °C for 30 min. The digested tissue was filtered through a 100 μm cell strainer (Becton Dickinson (BD) Biosciences, Franklin Lakes, NJ, USA) by adding 10 mL phosphate-buffered saline (PBS) including 10% Newborn Calf Serum (NCS, Sigma-Aldrich, St. Louis, MO, USA) and was finally centrifuged at 1000 rpm for 10 min. The supernatant was aspirated and the pellet, corresponding to the stromal vascular fraction, was washed once in 10 mL PBS/10% NCS, centrifuged again and resuspended in 10 mL PBS/10% NCS. Mesenchymal stem cells were split into three T25 flasks (Thermo Fisher Scientific, Waltham, MA, USA) and cultured at 37 °C in 5% CO_2_ and 95% air-humidified incubator. The complete cell culture medium constituted 45% Iscove’s Modified Dulbecco’s Medium (IMDM, Thermo Fisher Scientific, Waltham, MA, USA), 45% Gibco^®^ F-12 Nutrient Mixture (F-12, Thermo Fisher Scientific, Waltham, MA, USA), 10% HyClone Newborn Calf Serum (NCS, Sigma-Aldrich, St. Louis, MO, USA) and was supplemented with 0.1 mL Gibco^®^ Penicillin–Streptomycin (10,000 units/mL penicillin, 10,000 μg/mL streptomycin; Thermo Fisher Scientific, Waltham, MA, USA) and 10 µg basic fibroblast growth factor (recombinant human bFGF, Human Animal-Free recombinant), Millipore Merck, Darmstadt, Germany). At 80% confluency, cells were passaged. For further experiments, adipose-derived stem cells were used from the sixth passage. Cells were characterized by analyzing surface markers of stromal vascular fraction cells and adipose-derived stem cells via flow cytometry (BD™ FACS LSRII and FACSDiva™ software version 6.1.2, both BD Biosciences, Franklin Lakes, NJ, USA). Compensation was established using single stained controls and gating was performed with matched isotype/fluorescence minus one controls. In stromal vascular fraction samples, pericytes, supra-adventitial and endothelial progenitor cells below non-hematopoietic cells (CD45 negative) were identified [32]. Hematopoietic stem cells were excluded by staining against CD34, leucocytes by staining against CD45; CD106 (VCAM-1) staining was conducted in order to distinguish bone-marrow-derived stem cells from that of adipocytic origin. Adipose-derived stem cells were identified according to their positive expression of the in vitro surface markers CD105, CD13, CD73, CD90, and specific antigen for adipose-derived stem cells in the culture, CD36. All fluorochrome-conjugated antibodies and 7-aminoactinomycin D (7-AAD) dye to distinguish dead from viable cells were purchased from BD Biosciences (please see Appendix A).

### 2.2. Osteogenic Differentiation of Adipose-Derived Stem Cells

Osteogenic differentiation was initiated by supplementation of the growth medium with 2.5 mM dexamethasone, 50 μM L-ascorbate-2-phosphate, 100 μM vitamin D3 and 1 M ß-glycerophosphate. Adipose-derived stem cells expanded and maintained in the growth medium without supplements served as negative controls. Osteogenic differentiation was determined by Alizarin red S staining to visualize calcification, which was performed according to manufacturer’s protocol (Osteogenesis assay kit, #ECM815, Millipore Merck, Darmstadt, Germany). Matrix mineralization was determined by immunofluorescence labeling with anti-collagen I (ab34710), anti-sialoprotein (ab52128, Abcam, Cambridge, Great Britain) and anti-osteopontin (ab8448). As the secondary antibody Alexa 488-labeled secondary goat anti-rabbit antibody (1:100, Molecular Probes, Waltham, MA, USA) was used. After staining, cells were counterstained with 4′,6-Diamidin-2-phenylindol (DAPI) (Roche Diagnostics GmbH, Rotkreuz, Switzerland) for 15 min. Staining was investigated with an inverted confocal laser-scanning microscope (LSM780, Carl Zeiss, Jena Germany). Notably, images were taken at identical device settings to guarantee comparable results. The image processing was carried out using ZEN 2011 (Carl Zeiss Jena GmbH, Jena, Germany).

### 2.3. Electrical Stimulation System and Cell Preparation

The stimulation system has been described in detail in a previous study of our working group [26]. For cell experiments, adipose-derived stem cells were seeded onto glass coverslips with a diameter of 24 mm with a density of 2.2 × 10^4^ cells/cm^2^ and on electrodes with a density of 3.0 × 10^4^ cells/cm^2^ (Figure 1). After an adherence step for 30 min, chambers were covered with the lid, and cells were incubated over night at 37 °C under a 5% CO_2_ atmosphere.

### 2.4. Electrical Stimulation Pattern

Briefly, 24 h after cell seeding, adipose-derived stem cells were exposed to electrical stimulation with 1.7 V AC/20 Hz for 24 h, 72 h and 7 days, respectively. The stimulation pattern consisted of the continuous application of 1.7 V AC, 20 Hz using a Metric GX 305 function generator (Metrix Electronics, Hampshire, UK) set to 20 Hz. During stimulation, the cells were cultivated under standard cell culture conditions. Each test (cell number, cellular surface coverage, triphenyltetrazoliumchlorid (XTT) assay, cell cycle analysis) was conducted at a minimum of *n* = 6.

### 2.5. Cell Number and Cellular Surface Coverage

In order to visualize cell attachment on the electrodes after 1, 3 and 7 days of electrical stimulation, cells were stained with 1 μg/mL Calcein-acetoxymethyelster (Calcain-AM) (Thermo Fisher Scientific) diluted in fetal calf serum (FCS)-free medium. After 30 min of incubation at 37 °C and several washing steps with growth medium, micrographs were taken using FITC filters and 100-fold magnification (Axiovert 40 CFL, Carl Zeiss, Jena, Germany). Here, the number of attached cells was counted manually as described before [33]. Surface coverage of attached cells was quantified with ImageJ software (https://image.nih.gov/ij/) and expressed as the percentage of total area (each group at least *n* = 6) [34].

### 2.6. Cell Proliferation

Proliferation of adipose-derived stem cells was evaluated after 1, 3 and 7 days of electrical stimulation using an XTT assay according to the manufacturers’ manual (Cell Proliferation Kit II, Merck, Darmstadt, Germany). After 90 min of incubation, the optic density of the 96 well plates was evaluated using a Microplate Reader (Anthos 2010, Anthos Mikrosysteme, Krefeld, Germany) at a wavelength of 450 nm and reference of 630 nm as described in the literature [35].

### 2.7. Cell Cycle Analysis

Cell cycle analysis of stimulated and non-stimulated cells was carried out after 3 and 7 days using the 5-ethynyl-2′-deoxyuridine (EdU) assay in accordance with the manufacturers’ instructions (Click-iT™ EdU Alexa Fluor 488™ Flow Cytometry Assay Kit, cat. no. C10632, Thermo Fisher Scientific). In brief, adipose-derived stem cells were incubated with 10 µM EdU for 1 h. Cells of the same population without EdU staining served as a negative control. Moreover, in order to assess in which cell cycle phase proliferating cells were seen, FxCycle™ Violet Stain (cat. no. F10347, Thermo Fisher Scientific) was applied. Following incubation, the samples were washed in washing buffer containing 1% bovine serum albumin in phosphate buffer fixed using 2% paraformaldehyde and acquired using the flow cytometer device BD™ FACS LSRII equipped with fluorescence activated cell sorting (FACS) Diva™ software version 6.1.2 (both Becton Dickinson (BD) Biosciences, Franklin Lakes, NJ, USA).

Additionally, the extent of cell cycle progression and apoptosis (sub-G1 phase) in the cells was estimated by flow cytometric analysis after propidium iodide (Roche Diagnostics GmbH, Rotkreuz, Switzerland) staining. After treatment, cells were trypsinized with 0.05% trypsin ± 0.02% EDTA for 5 ± 10 min. The reaction was stopped with assay medium. Cells suspension was transferred to FACS tubes (Becton Dickinson (BD) Biosciences, Franklin Lakes, NJ, USA) and fixed in 70% ethanol for 12 or more hours at −20 °C. Briefly, after washing with PBS, cells were incubated with RNase (1 mg/mL) at 37 °C for 30 min. Finally, cells were re-suspended in propidium iodide (50 mg/mL) for at least 3 h at +2 to +8 °C protected from light until flow-cytometric analysis. The software FlowJo version 10.0.5 (FlowJo LLC, Becton Dickinson (BD) Biosciences, Franklin Lakes, NJ, USA) was used for data acquisition.

### 2.8. Statistics

Raw data sets were saved in Excel^®^ sheets (Microsoft Corporation, Redmond, WA, USA) and subsequently transferred into SPSS Statistics^®^ (version 23.0.0.2, MacOS X; SPSS Inc., IBM Corporation, Armonk, NY, USA). Data were expresses as means and standard deviations. Normal distribution was checked using the non-parametric Kolmogorov–Smirnov test and results were analyzed for statistical significance by the use of analysis of variance (ANOVA), unpaired non-parametric Mann–Whitney-U-tests, Wilcoxon Whitney tests and Student’s *t*-test. The level of statistical significance was set at *p* ≤ 0.05. Boxplots and bar charts were used for illustration purposes.

## 3. Results

### 3.1. Cell Characterization

Human adipose-derived stem cells that were passaged over time showed mesenchymal phenotype by high positive expression of CD105, CD13, CD73, CD90, and less than 3% or rare expression for CD45 and CD34 markers. Furthermore, flow cytometric data confirmed adipogenic origin of the cells by the expression of the CD36 antigen but very low expression (less than 0.5%) of CD106 marker (see Appendix A and Appendix A). Osteogenic differentiation of the adiopose-derived stem cells was monitored by Alizarin red staining to identify calcium deposits in a time series (Figure 2A). All isolated stem cells were able to differentiate into osteoblasts in a time range of 20–30 days. Immunohistological staining of collagen I, sialoprotein and osteopontin proved the expression of osteogensis markers after 20 days (Figure 2B).

### 3.2. Cell Number and Cellular Surface Coverage

After 24 h, no statistically significant differences in cell numbers or cellular surface coverages were detected when comparing stimulated and non-stimulated groups (both *p* > 0.05). In contrast, after 3 days, stimulated groups exhibited a significant increase—a 4.5-fold increase in cell numbers and a 2.7-fold increase of cellular surface coverage when compared to the non-stimulated group (both *p* < 0.001; Figure 3, Figure 4 and Figure 5).

After 7 days, a significant, 11-fold increase in cell numbers and a significant, 4-fold increase in cellular surface coverage were detected in the non-stimulated group when compared to electrically stimulated cells (both *p* < 0.001; Figure 4, Figure 5 and Figure 6).

### 3.3. Cell Proliferation/Relative Metabolic Activity

#### XTT

After 24 h, electrically stimulated cells revealed no differences regarding proliferation/relative metabolic activity values when compared to non-stimulated controls (*p* = 0.834, Figure 7). By contrast, after 3 days, a significantly higher proliferation was observed in stimulated cells (*p* < 0.001), whereas after 7 days, metabolic activity changed completely and values in the non-stimulated samples were significantly increased (*p* = 0.005; Figure 7).

### 3.4. Cell Cycle Analysis

Control cells harbored approximately 70% of the cells in the G0/G1 phase, 7.5% in S phase, and 20% in the G2/M phases, regardless of whether these cells were cultured for 3 or 7 days. In contrast, in the stimulated cells at day 3, cell cycle analysis revealed a shift towards the proliferative cell cycle phases (S and G2/M phase; Figure 8, Table 1). The sum of the proliferative phases increased from 28.1% of the non-stimulated cells to 50.5% of the stimulated cells, indicating a proliferative boost (Table 1).

At day 7, the stimulated cells were seen in 88.8% in the G0/G1 phase suggesting that the number of cells in the proliferative phase halved. Analyzing the sub-G1 phase, showing degraded DNA fragments as evidence for apoptosis initiation, a significant increase from 9.79% in control cells to 86.4% in stimulated cells was monitored (Figure 9).

## 4. Discussion

In the maxillofacial area, external electrical stimulation might be used to increase growth and proliferation of osteogenic stem cells in order to overcome xenogenous, alloplastic, allogeneic and autologous grafting procedures with their immanent limitations and risks [36,37,38,39]. Though, electrical stimulation of bone regeneration in this area has largely been unexplored and reported with inconsistent results [40,41,42]. This might be due to the fact that optimal parameters for electrical stimulation are not specified yet.

Tissue engineering strategies could be used in order to facilitate bone regeneration in cases of maxillofacial critical-size defects. For this issue, in particular, adipose-derived stem cells are especially suitable as they are abundant and functional with an osteogenic potential [43,44,45]. The cells used in the present experiment were extracted and characterized as described previously [30,31]. Nevertheless, as the cells were received from three different healthy donors, a heterogenous population can be assumed and cell response to electrical stimulation may differ with regard to the donor age, the location of harvesting as well as the harvesting method. For analysis of the osteogenic differentiation, staining with Alizarin red was successfully conducted as described in the literature [46].

In the present study, we demonstrated that continuous electrical stimulation significantly enhanced growth and proliferation of adipose-derived stem cells at the third day of stimulation. Furthermore, electrical stimulation led to a nearly doubled account of stimulated cells in S and G2/M phases when compared to controls. This early enhancement of stem cell growth and proliferation can be explained by cellular activation via asymmetric redistribution/diffusion of electrically charged cell membrane receptors in response to electric fields and via cell membrane depolarization due to direct activation of voltage-gated Ca^2+^ channels. In addition, the inverse effect of mechanotransduction (transformation of electrical stimulation into mechanical activity) causes tension in the cytoskeleton due to reorganization of cytoskeletal filaments and actin redistribution [22,47]. Further potential mechanisms include redistribution of cellular surface membrane proteins and lipids [48], stimulation of membrane-bound ATP synthesis [49] and activation of heat shock proteins [50].

However, at day 7, this beneficial effect was reversed. Hence, several potential negative effects of continuous electrical cell stimulation have to be considered. Firstly, there might be a certain amount of unwanted corrosion at the titanium electrodes in stimulated samples, leading to decreased cell growth and proliferation. For this reason, some authors recommended platinum for electrode material even if it is inferior in stiffness when compared to other metallic materials and considerably more expensive [51]. Secondly, the continuous electrical stimulation that was used in the present experiment might have led to excessive accumulation of charged proteins out of the supernatant at the electrode/cell interface that might have resulted in exceeding cell damage [21,52]. Continuous electrical stimulation has been reported to create intermediates such as hydrogen peroxide, hydroxyl and oxygen ions [53] and leads to an increased medium pH as well [54] that might be toxic for stem cells. Though, analogous to the present results, Kim et al. evidenced that continuous electrical stimulation of osteoblasts at day 2 and 4 was more favorable for osteoblast proliferation when compared to pulsatile stimulation [21]. However, for later time points, beneficial results were reported using pulsatile electrical stimulation in terms of increase in osteogenic differentiation, increased bone formation [55,56,57,58], and in terms of a modified macrophage response [59]. In accordance, further research is needed for comparing continuous and pulsatile electrical stimulation in in vitro and in vivo models.

From the findings of the present experiment and the current references, we hypothesize, that an initial continuous electrical stimulation in combination with a pulsatile stimulation at later time points may yield the most beneficial effect of cell growth as well as proliferation of adipose-derived stem cells.

## Figures and Tables

**Figure 1 biomedicines-08-00482-f001:**
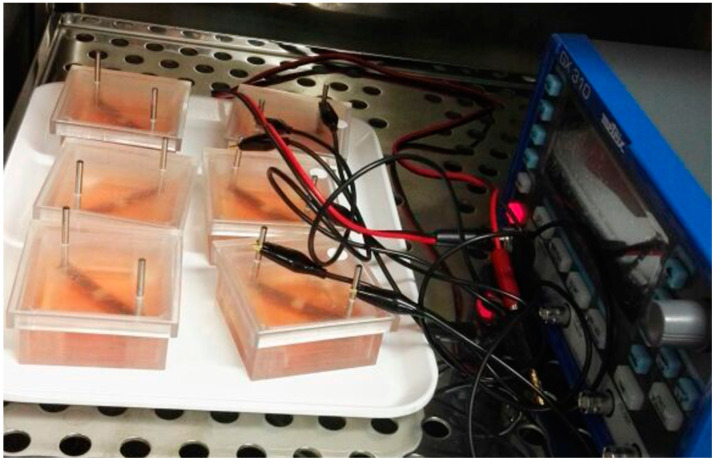
Experimental setup showing stimulated and non-stimulated adipose-derived stem cells seeded on coverslips with electrodes.

**Figure 2 biomedicines-08-00482-f002:**
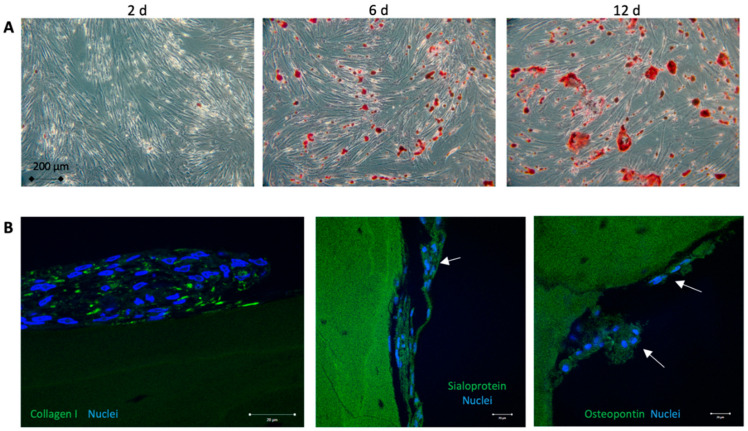
Proof of osteogenic differentiation. (**A**): Osteogenic differentiation of the adiopose-derived stem cells was monitored by Alizarin red staining to identify calcium deposits in a time series. All isolated stem cells were able to differentiate into osteoblasts in time range of 20–30 days. (**B**): Immunohistological staining of the osteogenesis markers collagen I, sialoprotein and osteopontin (green fluorescence) counterstained with Hoechst (blue) to label the nuclei of the adiopose-derived stem cells after 2 weeks. In the undifferentiated control group, no expression markers were seen.

**Figure 3 biomedicines-08-00482-f003:**
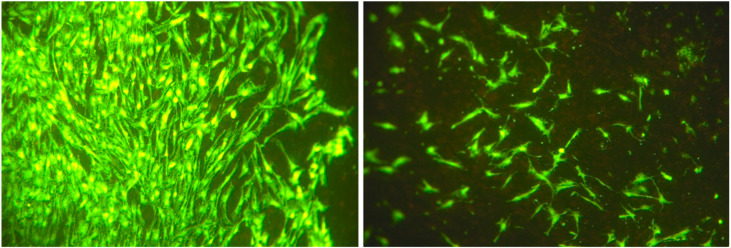
Evaluation of cell number and cellular surface coverage of adipose-derived stem cells electrically stimulated (**left**) and non-stimulated (**right**) at day 3 (original magnification ×100).

**Figure 4 biomedicines-08-00482-f004:**
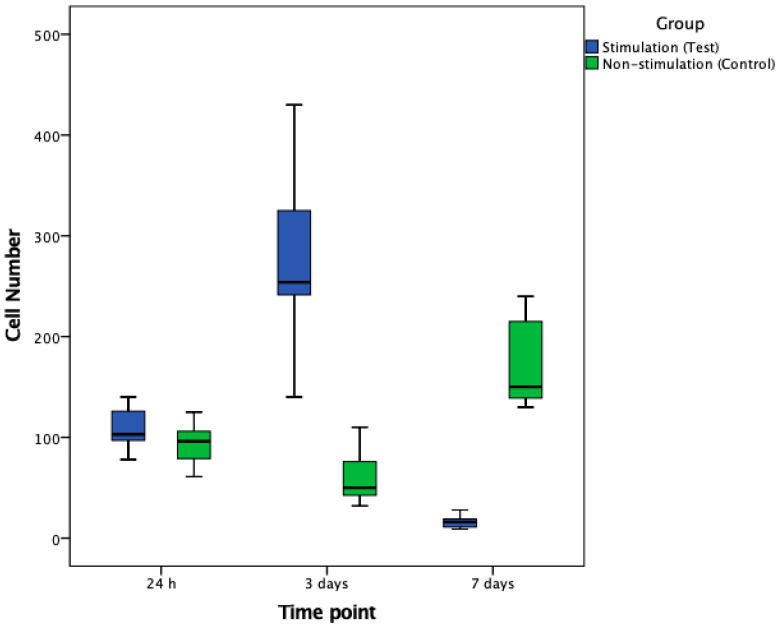
Boxplots for illustration of cell numbers in stimulated and non-stimulated cell samples after 24 h, 3 and 7 days of continuous electrical stimulation. At days 3 and 7, significant differences were seen (each *p* < 0.001).

**Figure 5 biomedicines-08-00482-f005:**
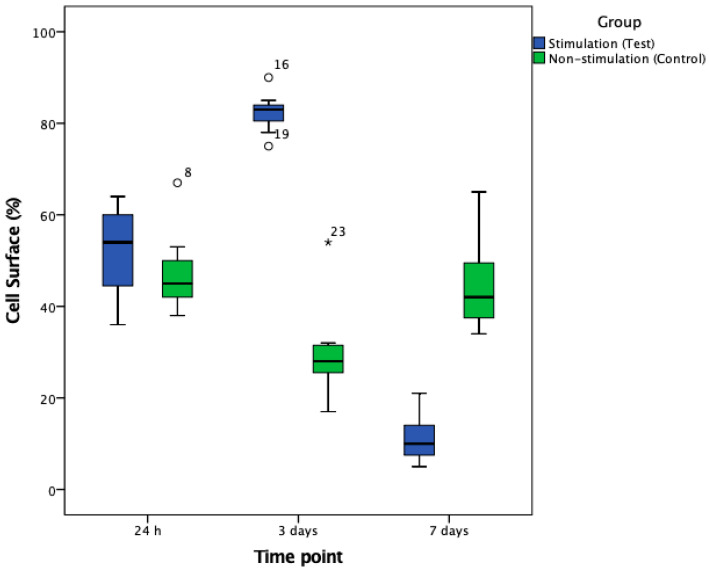
Boxplots for illustration of cellular surface coverage in stimulated and non-stimulated cell samples after 24 h, 3 and 7 days of continuous electrical stimulation. At days 3 and 7, significant differences were seen (each *p* < 0.001).

**Figure 6 biomedicines-08-00482-f006:**
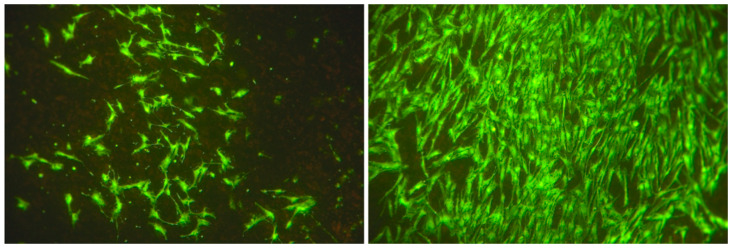
Evaluation of cell number and cellular surface coverage of adipose-derived stem cells electrically stimulated (4A) and non-stimulated (4B) at day 7 (original magnification ×100).

**Figure 7 biomedicines-08-00482-f007:**
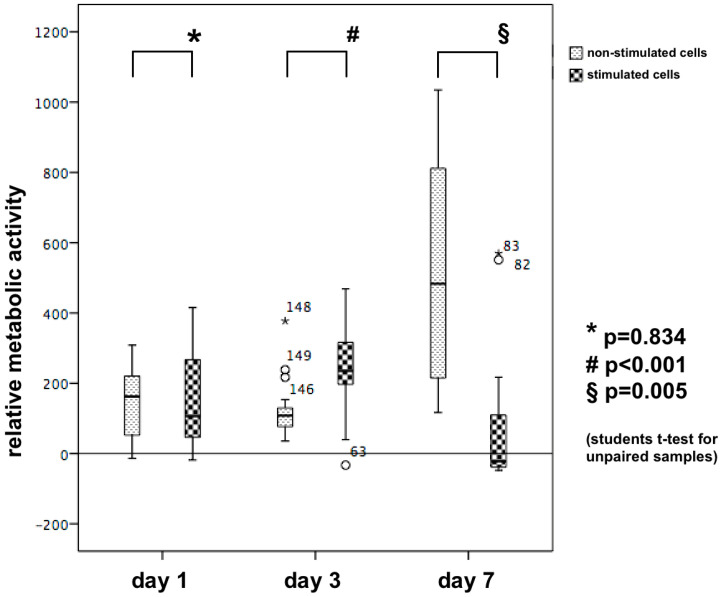
Boxplots for illustration of the proliferation/relative metabolic activity in stimulated and non-stimulated cell samples after 24 h, 3 and 7 days of continuous electrical stimulation. At day 3, the proliferation was doubled in the stimulated cells and in contrast, at day 7 this was doubled in the controls while the stimulated cells reduced their proliferation rate.

**Figure 8 biomedicines-08-00482-f008:**
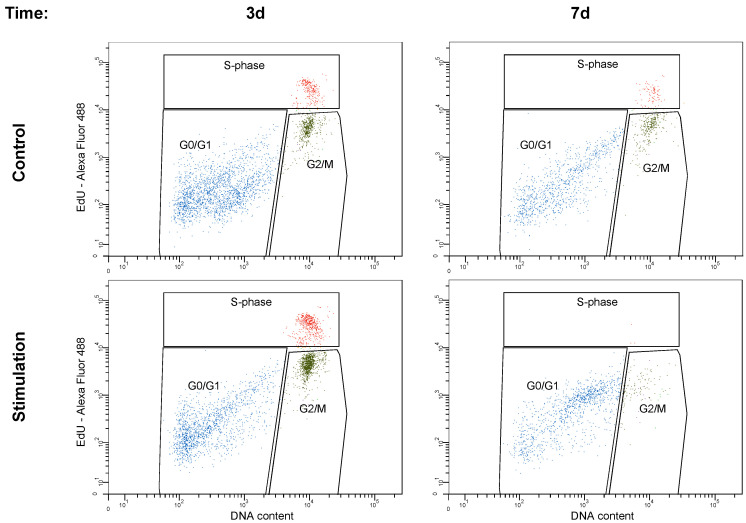
Histograms illustrating differences in the cell cycle phases after 3 and 7 days for stimulated and non-stimulated (control) cells.

**Figure 9 biomedicines-08-00482-f009:**
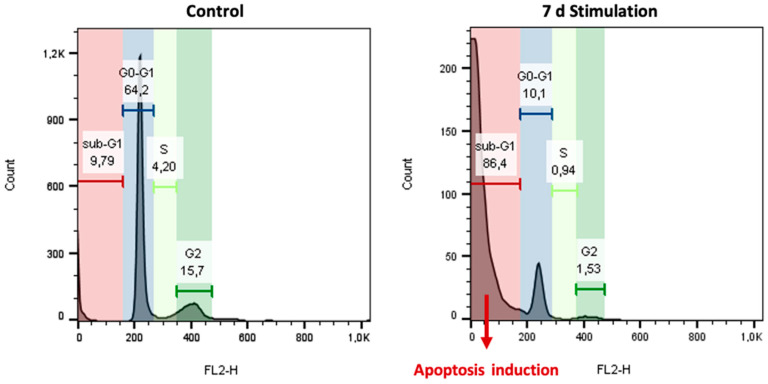
Analysis of the sub-G1 phase for the measurement of apoptotic cell via flow cytometry. After 7 days of electric stimulation, cell cycle analysis revealed a significant increase in the apoptotic cells.

**Table 1 biomedicines-08-00482-t001:** Distribution of cells within three phases of the cell cycle. Here, after three days, electrical stimulation led to a higher shift of stimulated cells (“stimulation”) in S and G2/M phases when compared to non-stimulated controls (“control”). After seven days of electrical stimulation, an adverse effect was seen.

Time		G0/G1	S	G2/M
3 d	Control	71.1 ± 6.8	8.4 ± 3.1	19.7 ± 5.7
	Stimulation	48.6 ± 8.8	15.3 ± 6.7	35.2 ± 9.6
7 d	Control	70.1 ± 5.8	6.5 ± 4.8	22.0 ± 4.7
	Stimulation	88.8 ± 7.2	0.3 ± 2.3	10.4 ± 5.9

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
