# Peer review of "Continuous Electrical Stimulation Affects Initial Growth and Proliferation of Adipose-Derived Stem Cells"

_biomedicines, 2020, doi:10.3390/biomedicines8110482_

Round 1

Reviewer 1 Report

The manuscript is a resubmission and has been corrected for concerns raised by the reviewer in the initial version. The authors provide more quantitative data, which improves the merit of the paper. There are some points, which still need attention. 

In Fig. 2B, the undifferentiated control is missing. If there is no expression of markers in undifferentiated controls, this should be mentioned in the text at least.

It seems that histograms in Fig. 8 do not reflect the figures as shown in table; esp. at day 3, S and G2/M.

Line 283/284: This statement is only true for values at day 3, not day 7.

Line 302: “Multipotency” has not been tested here. Please avoid this term.

The English style has to be improved. Following here only a few examples

Line 200: Cells isolated from the fatty tissue that were passaged over time

Line 211: Evidence of

Line 270: remained to

Line 300: assessable

Line 328: analogue

Author Response

Reviewer 1:

Comment #1:

“In Fig. 2B, the undifferentiated control is missing. If there is no expression of markers in undifferentiated controls, this should be mentioned in the text at least.“

Our response:

This is correct. In the undifferentiated controls, no expression markers were seen. This was added to the text.

Comment #2:

“It seems that histograms in Fig. 8 do not reflect the figures as shown in table; esp. at day 3, S and G2/M.”

Our response:

You are absolutely right! We are very sorry for this mistake and included the correct figure.

Comment #3:

“Line 283/284: This statement is only true for values at day 3, not day 7.”

Our response:

This is true and we corrected the statement.

Comment #4:

“Line 302: “Multipotency” has not been tested here. Please avoid this term.“

Our response:

We agree and omitted the term.

Comment #5:

“The English style has to be improved. Following here only a few examples.”

Our response:

We tried our best and conducted another proof-reading by a native speaker.

Comment #6:

“Line 200: Cells isolated from the fatty tissue that were passaged over time; Line 211: Evidence of; Line 270: remained to; Line 300: assessable; Line 328: analogue.”

Our response:

Changes were done.

Reviewer 2 Report

The authors improved the manuscript as per my comments. However, some of the points should be reconsidered again:

  1. The Abstract should be precise and does not carry all the results, rather a brief emphasis.
  2. In Keywords: Delete 'stem cells' and use 'human adipose-derived stem cells', delete 'growth', and 'viability'.
  3. Replace the Fig. 3, the present image has extensive background fluorescence. I don't think it's a publication-quality image. Similar problem with Fig. 6.

Author Response

Reviewer 2:

Comment #1:

“The Abstract should be precise and does not carry all the results, rather a brief emphasis.“

Our response:

We are not sure how to shorten the abstract any further. Minor changes were made.

Comment #2:

“In Keywords: Delete 'stem cells' and use 'human adipose-derived stem cells', delete 'growth', and 'viability'.”

Our response:

Done

Comment #3:

“Replace the Fig. 3, the present image has extensive background fluorescence. I don't think it's a publication-quality image. Similar problem with Fig. 6.”

Our response:

We agree but we could not get better figures out of our microscope as the slight background fluorescence was persisting. If needed, we could modify the pictures using photoshop but we would rather like to keep the original ones, if possible.  

Round 2

Reviewer 1 Report

Fine for me now.

This manuscript is a resubmission of an earlier submission. The following is a list of the peer review reports and author responses from that submission.

Round 1

Reviewer 1 Report

The paper Dau and colleagues describes the beneficial effect of electrical stimulation on the growth and viability of hADSC. The paper deals with an interesting topic, i.e., propagation of stem cells for downstream use in autologous cell transplantation therapies. The authors used undifferentiated cells without testing their functional capacities in terms of osteogenic differentiation in vitro and in vivo. Additionally, it is unclear, whether electrical stimulation is responsible for the increase of growth, or whether electrical stimulation increased e.g., the secretion of factor(s), which in an autocrine manner may increase proliferation. Without these functional inveatigations the data are merely descriptive and preliminary at this point.

Line 266. The authors state that featured regular metabolic activity and proliferation. The XTT assay rather reflects proliferation than metabolic activity of single cells.

Line 93: Use subscript for CO2

Line 176: expressed

Line 195: Include also reference to Figure 4, because Figure 2 does not show quantitative data. It is also unclear, whether the figures mentioned (6-, resp., 3-fold) refer to the cell number or to the surface coverage. Quantitative data on surface coverage are not shown. Same for Figure 3 and 4.

Figure 4: Initial cell numbers should be included in order to demonstrate increase or decrease in cell numbers.

Line 214-222: Values should be shown as a plot. Its hard to follow the text. As far as I see, the difference between 24 h and 3 days of stimulation is not very obvious. Section 3.3. does not show anything on proliferation; please change heading accordingly.

Figure 5: Please explain “bright” and “low” to the non-informed reader and explain their functional meaning.

Line 238: “arrest” does not seem the correct term. Cell cycle arrest/progression has not been adressed. How is the situation at day 7?

Reviewer 2 Report

The manuscript titled "Continuous electrical stimulation affects initial growth and vaibility of adipose derived stem cells" is well-written and described the effect of electric field stimulation on growth and differentiation of hADSCs. However, the contents of the manuscript is not novel since, a lot of papers publised regarding the EFs stimulation. Therefore, the author should mention the importance of this study in details throughout the manuscript and carefully justify the results.